# How Do Religiosity and Spirituality Associate with Health-Related Outcomes of Adolescents with Chronic Illnesses? A Scoping Review

**DOI:** 10.3390/ijerph192013172

**Published:** 2022-10-13

**Authors:** Nicolò M. Iannello, Cristiano Inguglia, Fabiola Silletti, Paolo Albiero, Rosalinda Cassibba, Alida Lo Coco, Pasquale Musso

**Affiliations:** 1Department of Law, University of Palermo, 90134 Palermo, Italy; 2Department of Psychology, Educational Science and Human Movement, University of Palermo, 90128 Palermo, Italy; 3Department of Educational Sciences, Psychology, Communication, University of Studies of Bari, 70122 Bari, Italy; 4Department of Developmental Psychology and Socialisation, University of Padua, 35121 Padova, Italy

**Keywords:** religiosity, spirituality, adolescence, chronic illness, adjustment

## Abstract

The aim of the current scoping review was to explore the associations between religious and spiritual factors and the health-related outcomes of adolescents with chronic illnesses, as well as to investigate possible mechanisms accounting for these relationships. In total, 20 studies meeting the eligibility criteria were reviewed after performing a search of the Scopus, Web of Science, and PubMed databases. The results suggested that religious and spiritual beliefs, thoughts, and practices (e.g., spiritual coping activities) might have both beneficial and deleterious effects on the way adolescents deal with their medical condition, on their psychosocial adjustment, on their mental and physical health, and on their adherence to treatments. Mediating and moderating mechanisms explaining these relations were also evidenced. Suggestions for future research and practical implications for healthcare professionals are provided in the concluding section of this work.

## 1. Introduction

Adolescence is a pivotal time of life, characterized by a number of developmental tasks, such as dealing with body transformations, the formation of a clear identity and a sense of autonomy and confidence, and the establishment of romantic and peer relationships (for review, [1,2]). One factor that might hinder this development is chronic disease [3]. Indeed, such a condition might impose lifestyles on adolescents that isolate them (e.g., repeated hospitalization), might make them unsatisfied with their bodies, which might be misshaped by surgical scars, and might prolong their dependence on parents who are in charge of their medical treatments [4]. Thus, chronic diseases might expose adolescents to several psychological, emotional, and relational struggles affecting their adjustment, health, and quality of life [5]. Understanding the factors that might help adolescents with chronic illnesses adapt to the consequences of their condition is both theoretically relevant and helpful for the implementation of proper intervention programs [6]. 

Among psychological factors, religiosity and spirituality might be regarded as good candidates for this scope as they provide adolescents with several cognitive (e.g., worldviews) and social resources (e.g., connection with religious groups; [7]) that might help them to find comfort, hope, and purpose in the experience of illness [8]. Nevertheless, the number of studies investigating the role of religiosity and spirituality in the lives of adolescents diagnosed with a chronic disease is still sparse and limited [9]. Recently, there have been a few attempts to systematize these contributions. However, such literature [9,10] or meta-analytic reviews [11] and qualitative meta-syntheses [12] did not explicitly clarify potential explanatory mechanisms connecting religiosity and spirituality with the health-related outcomes of adolescents with chronic illnesses. In addition, such studies prevalently focused on specific chronic diseases (e.g., sickle cell disease; [10]), took into account only specific aspects of religious and spiritual experience (e.g., religious/spiritual coping strategies; [11]), or involved different age groups from childhood to young adulthood (e.g., [12]). In our view, albeit illuminating, these contributions might hinder the possibility (1) to shed light on how religious and spiritual dimensions affect the health-related outcomes of adolescents with different types of chronic diseases, (2) to unveil other aspects of adolescents’ religious and spiritual experiences (e.g., religious commitment) that might help them to deal with being ill, and (3) to uniquely tap into the developmental phase of adolescence. Thus, this scoping review has the broader goal of identifying current knowledge that investigates why and how adolescents turn to their religious and spiritual dimensions (i.e., thoughts, beliefs, practices, and communities) to go through various chronic medical conditions that differ in severity, nature, and implications for young patients. Before reviewing the related research, we first conceptualized religiosity, spirituality, and chronicity, and then demonstrated the relevance of connecting spirituality, religiosity, and chronic illness during adolescence.

### 1.1. Defining Religiosity and Spirituality

When addressing religious and spiritual issues, one of the main concerns is to provide a clear conceptualization of religiosity and spirituality and to determine whether they are related or separate constructs [13]. Broadly speaking, religiosity might be defined as the public or private adherence to beliefs, rituals, and practices of religion [14], whereas spirituality can be defined as the individual search for a meaning in life, for a personal connection with transcendent realities (God/High Power), and for interconnectedness with something greater than the self (e.g., humankind, the whole of creation; [15]). 

With regard to their reciprocal relations, religiosity and spirituality have been often regarded as different, but overlapping constructs [14] sharing reference to the sacred [16], which encompasses the divine, the transcendent, or any aspect of life that individuals perceive as imbued with a spiritual/divine character [17]. Interestingly, spirituality has been considered as a phenomenon pertaining to all individuals, regardless of their religious affiliation [18], and religiosity as one of the possible contexts where one’s spiritual inclinations might be expressed [19,20]. In any case, it is worth noting that, in their lived experiences of religiosity and spirituality, individuals might self-describe as spiritual and religious, spiritual but not religious, religious but not spiritual, or neither religious nor spiritual, according to the meanings attributed to these terms [21]. Finally, a controversial aspect of religiosity and spirituality concerns the role in adolescents’ development, with some works showing unique and others joint influences of these two dimensions on their outcomes (e.g., promoting mental health, discouraging risky behaviors; for review, [22]).

As a result, the ongoing academic debate on the definitions of religiosity and spirituality constructs, their complex interrelations, and their exclusive or shared impacts on adolescents’ growth are not clear or definitive. In view of this, in the current scoping review, we adopted an explorative approach by considering individuals’ identifications with religious or spiritual aspects, or both [23]. 

### 1.2. Conceptualizing Chronicity

Intuitively, the term *chronic* identifies a continuous medical condition [24], but scholars have widely debated on how long such a condition has to last to be considered as permanent or persistent [25]. For this, the time intervals often adopted to categorize a disease as chronic are 3, 6, or 12 months (for review, [26]), and also differ according to the type of disorder [27]. In addition to duration, *chronic* also refers to recurrence, another feature that characterizes and distinguishes an ongoing disease from an acute disease [26]. Indeed, an acute disease has a specific cycle—onset, a manifestation of signs/symptoms, and recovery—whereas a chronic one continues endlessly, is insidious, worsens, or remains in remission for a long time [28]. Briefly, acute diseases might be curable, while chronic ones are not [27]. 

According to the World Health Organization [24], chronic conditions encompass: *mental, neurological, and substance-use disorders* (e.g., depression, schizophrenia, and epilepsy); *non-communicable diseases*, such as cardiovascular disease, diabetes, cancer, chronic respiratory disorders, and sensory, digestive and musculoskeletal disorders; and *communicable diseases*, such as HIV/AIDS, tuberculosis, and viral hepatitis. However, beyond any possible and exhaustive taxonomy, it should be borne in mind that a chronic disease significantly impacts patients’ qualities of life, as it might impair people’s functions, daily activities and social roles; might increase their need for medical care and use of medical technology; and might impose on them accommodations and frequent assistance from others and carers [25]. 

In the current scoping review, a broad and general conceptualization of chronicity was considered—one that does not focus on any standard duration and applies to the different types of chronic conditions. Additionally, a distinction between disease and illness has been taken into account, given that, as it has been suggested, disease “refers to the pathophysiology of a condition”, whereas illness “is the human experience of a disease” [29] (p. 4). In our view, it is at the latter level that individuals’ religious and spiritual strategies to cope and adjust might be investigated. Based on these premises, the following definition is adopted:

“Chronic illness is the irreversible presence, accumulation, or latency of disease states or impairments that involve the total human environment for supportive care and self-care, maintenance of function, and prevention of further disability” [30], as cited in ([28], p. 8).

### 1.3. Adolescence, Chronic Illness, Religiosity, and Spirituality 

Adolescents diagnosed with a chronic medical condition are confronted with situations and experiences (e.g., invasive treatments) that might alter the normal processes of growth at different levels, according to the disease severity and implications for young patients [31,32]. In detail, at the biological level, a chronic condition might delay puberty; at the psychological one, it might impair adolescents’ self-esteem and self-image, as well as impact their mental health outcomes. Additionally, it might have consequences on adolescents’ social life, as it might increase their dependence on parents, isolate them from peers, and interfere with their intimate relationships. Finally, it might negatively affect adolescents’ school attendance and attainment of educational and academic goals (for review, [4]). 

Concurrently, adolescents’ growth might, in turn, impact the course of chronic diseases. In other words, adolescents’ typical developmental needs, such as feeling accepted by others, being involved in peers’ activities, and exploring new roles and behaviors, might conflict with the lifestyles that a chronic disease might impose on young patients, who may pay less attention to illness treatment and management in order to satisfy other priorities [4,33]. In summary, adolescents growing up with a chronic disease must face several stressors (e.g., restrictions in their social life) and psychological issues (e.g., frustration and illness acceptance; [34]).

Religiosity and spirituality may contribute to the psychosocial functioning and well-being of adolescents with chronic illnesses for both general and specific reasons. First, theoretical considerations and empirical evidence, also related to the developmental advancements of adolescents (e.g., increased capacity of abstract thinking), showed that religiosity and spirituality may generally provide adolescents with ideological, social, and transcendent contexts offering them perspectives, webs of support, and a sense of connectedness with something greater that might help them to be aware of their uniqueness and find meaning and purpose [35]. Second, as adolescents with chronic illnesses are more often confronted with uncertainty about the future, finitude of life, psychological burdens, and negative experiences [31], religious/spiritual beliefs and values may specifically help them to find meaning in life and answers to crucial existential questions. 

Interestingly, despite the well-documented influences of religiosity and spirituality on adolescents’ mental/physical health (for review, [36]) and development (for review, [37]), the list of studies investigating their role in the lives of adolescents with a chronic disease is still short [9]. Additionally, questions regarding the explanatory processes underlying the associations between these two constructs and the health-related outcomes of adolescents with chronic illnesses have been only marginally addressed.

### 1.4. The Current Study

Based on the above considerations, in the current work, we were interested in mapping and collecting different types of evidence analyzing how religiosity and spirituality might be related to the various health-related outcomes (e.g., adjustment, adherence to treatments, coping, mental and physical health, meaning and purpose, and life quality) of adolescents with chronic illnesses and in uncovering the explanatory mechanisms accounting for these linkages. In detail, we aimed to: (1) assess the extent of the body of literature on this topic; (2) understand how research is conducted in this field; and (3) detect and examine knowledge gaps [38]. 

With these being the main goals, we opted for a scoping review approach, as scoping studies are aimed at identifying the current state of understanding by contextualizing knowledge and highlighting what is currently known by scholars [39]. To note, in the process of carrying out this scoping review, we decided to focus on the adolescent years ranging from 10 to 20 in order to tap into different phases of this developmental period, namely early (10–13), middle (14–17), and late (17–20) adolescence [31]. This choice was linked to the assumptions that chronic illness [31], religiosity, and spirituality [40] might be uniquely experienced according to differences in biological, cognitive, and social development related to the various phases of adolescence.

## 2. Method

### 2.1. Selection of studies

Studies for this scoping review were identified in two ways. First of all, the electronic *Scopus*, *Web of Science*, and *PubMed* databases were queried. The following search terms were used: (“chronic illness” OR “chronic disease”) AND (spirit* OR religio*) AND (adolescen* OR youth*). Titles, abstracts, and keywords were the research fields when searching the *Scopus* and *Web of Science* databases, while the corresponding *PubMed* research field included only titles and abstracts. However, for the terms (adolescen* OR youth*), we extended the search to all research fields. For the *Scopus* database, due to the high number of results inappropriate for the purpose of this work, we selected the following subject areas: Psychology, Social Sciences, Arts and Humanities, Health Professions, and Neuroscience. In addition, a further search was carried out by hand-searching the reference lists of the eligible papers (see below). The search included no time span constraints and any geographic region. 

To be included in the scoping review, a study (a) had to be published in full in English or Italian; (b) had to be conducted, although not exclusively, with adolescents aged 10 to 20 years with a chronic illness; (c) had to measure religious and/or spiritual dimensions; and (d) had to report information on at least one adolescent’s health-related outcome. Book chapters and review papers were excluded, as well as measure validation and pilot studies. Figure 1 shows the Preferred Reporting Items for Systematic Reviews and Meta-Analyses (PRISMA) flow diagram, detailing study selection. The initial search resulted in 443 records. After removing 81 duplicates, the titles and abstracts of 362 manuscripts were screened. Screening identified 106 studies appearing to meet the inclusion criteria. After reading the full texts of these eligible studies, 20 fully met the inclusion criteria, including three new studies found by hand-searching the reference lists.

### 2.2. Classification of Studies

Living with a chronic disease is a complex process causing adolescents to struggle with the desire to have a normal life. Religiosity and spirituality have generally been found to contribute to individuals’ chronic illness experiences both positively and negatively [9,41]. Therefore, the studies of the current scoping review were grouped according to the beneficial or deleterious relationships of religious/spiritual beliefs, thoughts, and practices and young patients’ health-related outcomes, such as the ability to cope, psychosocial adjustment, mental/physical health, quality of life, and adherence to treatment.

## 3. Results

As mentioned above, on the basis of the inclusion/exclusion criteria, 20 studies were included in this scoping review, whose main characteristics are reported in Table 1. In general, 6 studies were qualitative and 14 were quantitative (5 had a longitudinal design). The studies focused on a variety of chronic diseases, such as congenital heart disease, HIV, sickle cell disease, cystic fibrosis, inflammatory bowel disease, Type 1 diabetes, cancer, asthma, and spina bifida. In the following paragraphs, their main findings are described.

### 3.1. Positive Relationships of Religiosity and Spirituality with Health-Related Outcomes of Adolescents with Chronic Illnesses 

As a whole, the studies in the current scoping review highlighted that religiosity and spirituality might be regarded as potential psychological factors sustaining adolescents in their experience of chronic illness. In particular, several works revealed that young patients turned to their faith, religious/spiritual beliefs, practices, and communities *to cope and find strength, meaning, purpose, comfort, and hope*. In this regard, Atkin and Ahmad [42], by means of in-depth interviews among adolescents diagnosed with thalassemia major or sickle cell disorder, demonstrated that those from Muslim backgrounds came to terms with and made sense of their condition because of their faith in Allah and regular prayers. Interestingly, this process seemed to be more evident among older adolescent boys, who more actively explored Islam than the more passively adherent to their families’ religious observances in their younger counterparts. Conversely, Christian participants stated that their religion was not equally relevant, although some respondents admitted to drawing strength from God. To note, among Christians, those under twelve were more inclined to consider their religion as a source of relief than those over sixteen, who did not see religion as an important coping strategy. Beyond age, gender, and religious affiliation, other demographic characteristics seemed to relate to the use of religion/spirituality as a coping strategy among pediatric patients. Indeed, as reported by Landolt and colleagues [53], injured and children newly diagnosed with a chronic disease who had a lower socioeconomic status adopted religious coping more often. Additionally, Cotton and colleagues [46] found that, among urban adolescents with asthma, African American adolescents scored higher on religious/spiritual measures, including positive religious coping, than non-African American peers, highlighting the importance of taking ethnicity into consideration.

Alvarenga and colleagues [43], in their exploration of the spiritual needs of children and adolescents with cancer, cystic fibrosis, and type 1 diabetes mellitus, by means of interviews with photo-elicitation, found that believing in a benevolent God gave pediatric patients hope and helped them to integrate the meaning of their disease. Encouragement and comfort were also gained from members of their religious communities. In line with that, Silveira and Neves [58], by means of semi-structured interviews among children and adolescents who needed special health care services from South Brazil, demonstrated that churches were relevant components of ill teens’ social networks as they received from them spiritual and emotional support. Another qualitative study [60] on the spiritual needs of adolescents with different types of chronic diseases from Thailand and from Buddhist backgrounds reported similar findings. In particular, this study highlighted that those young patients referred to their families and parents as spiritual anchors from whom they obtained spiritual support when they felt pain or bad, and strength to fight against the illness. In addition, it showed that religious practices (e.g., meditating, praying, and making merit at the temple) increased because of adolescents’ conditions and that, from those rituals, adolescents gained inner peace, spiritual contentment, and relief. 

Religion has been found to facilitate the management of congenital heart disease among adolescents and their parents from Palestine, who stated that illness is God’s will and that, as a consequence, it should be tolerated, rather than objected; religious activities were, thus, intended as a means to receive help, hope, and healing [49]. Pediatric patients with HIV also endorsed feeling God’s presence and being part of a larger force to a greater extent than their HIV-negative counterparts [44]. In the same way, adolescents living with sickle cell disease defined religiosity and spirituality as strategies for overcoming difficulties related to their health issues. Along this line, Clayton-Jones and colleagues’ qualitative study [45] highlighted that religiosity and spirituality are beneficial in different ways. Indeed, young patients (with sickle cell disease) from a Christian background described spirituality as a means to enhance their sense of connectedness with others, with nature, and with arts, which, in turn, helped them feel restored, find meaning, and transcend their condition. They saw religiosity, instead, in terms of a connection with God that might be reinforced through prayers, participation in religious services, and traditions. Additionally, they considered reading the Bible as a source of direction and help. Briefly, it was through connection with God and scriptural metanarratives that adolescents seemed to gain new outlooks on their illness, strength, hope for the future, and frameworks for decision-making and reflection. 

The religious and spiritual dimensions of adolescents with chronic illnesses also *favor their mental health, psychosocial adjustment, and quality of life*. Considering this, data from the Lyon and colleagues’ study [54], which were collected as part of two-site, two-armed randomized controlled clinical trial, among primarily African American adolescents diagnosed with HIV revealed that spiritual well-being (faith and sense of meaning/peace) was associated with lower depression and anxiety, and greater life quality. Akin to this, Zehnder and colleagues [61] reported beneficial effects of religion on pediatric samples’ psychological adjustment. In particular, scholars reported that religious coping (asking for God’s help and praying to God for comfort) reduced post-traumatic stress symptoms among injured and adolescents newly diagnosed with a chronic disease over time. 

In a more nuanced way, Cotton and colleagues [47] explored whether the perception of one’s relationship with God/High Power (*religious well-being*) and the degree of satisfaction and meaning/purpose in life (*existential well-being*) impacted adolescents’ mental health differently and whether disease status (inflammatory bowel disease vs. healthy peers) moderated such an association. Interestingly, scholars found that existential well-being was positively related to emotional functioning and the inverse relationship between religious well-being and depressive symptoms was stronger for adolescents with inflammatory bowel disease than for their healthy counterparts. Thus, these findings highlight the greater relevance of spiritual issues in the mental health of adolescents with a chronic illness. 

One study [56] investigated the prospective relationships (two time points about 2 years apart) between spiritual coping and psychological adjustment among adolescents with chronic diseases (cystic fibrosis and diabetes). The results indicated that, regardless of disease, positive spiritual coping (seeking spiritual support from God and benevolent religious reappraisals) predicted fewer depressive symptoms over time. Interestingly, this study demonstrated that positive spiritual coping also discouraged maladaptive coping strategies, such as negative spiritual coping (feeling punished and/or abandoned by God). In another study, Reynolds and colleagues [55] sought to identify the explanatory mechanisms of the association between spiritual coping and psychosocial adjustment in the same sample of adolescents. Specifically, they revealed that, regardless of the patients’ ages (younger vs. older adolescents), positive spiritual coping was associated with fewer internalizing (depression and anxiety) and externalizing problems (hyperactivity, aggression, and conduct problems) by means of an optimistic attributional style (adolescents’ positive appraisal of disease-related events). However, multi-group analyses revealed that positive spiritual coping was associated with more optimistic attributions only among adolescents with diabetes. The optimistic attributional style, thus, mediated the effect of positive spiritual coping on psychosocial adjustment (internalizing and externalizing problems) only among adolescents with diabetes. Akin to this, Grossoehme and colleagues [50] explored whether a sense of meaning/peace and faith explained the associations between different religious/spiritual factors and psychological adjustment among young people between 14 and 21 years of age with cancer. In detail, scholars found that feeling God’s presence and considering oneself very religious were negatively and indirectly associated with anxiety, depressive symptoms, and fatigue through a higher sense of meaning and peace.

Research has also emerged supporting the role of religiosity and spirituality in the *physical health* outcomes of adolescents with chronic illnesses. Regarding this, Clayton-Jones and colleagues’ qualitative study [45] demonstrated that, for young patients, following religious principles and having a relationship with God were important to focus on and improve their health. In addition, in their 5-year longitudinal investigation on spiritual coping and health functioning among adolescents with cystic fibrosis, Reynolds and colleagues [57] discovered that positive spiritual coping predicted a slower decline in pulmonary function and nutritional status, as well as fewer days of hospitalization. These findings contrast with those reported by D’Angelo and colleagues [48], who did not find any longitudinal association between positive/negative religious/spiritual coping and health functioning among adolescents with the same pulmonary condition. However, interestingly, scholars revealed that frequent hospitalization attenuated the use of negative religious/spiritual coping concurrently at follow-up.

*Adherence to medical treatment* is another religiosity/spirituality-related positive association that emerged from the reviewed studies. For instance, Atkin and Ahmad [42] evidenced that the belief in the obligation to do the best for oneself pushed older Muslim adolescents to adhere to medical treatments. In addition, religious communities emerged as relevant sources encouraging young patients to not stop exercising their care [58]. Grossoehme and colleagues [51], by using the theory of reasoned action, elucidated the possible psychological processes underlying the association between spiritual coping and engagement with airway clearance among adolescents with cystic fibrosis. In detail, scholars reported that engaged spirituality (a factor including positive religious coping, collaboration with God to solve problems, turning problems to God, and viewing one’s body as sacred) may promote positive perceptions of treatment utility and compliance with subjectively perceived social norms by close friends for treatment completion; in turn, both of the latter dimensions were associated with stronger intentions to perform the therapy and to adhere to the treatment. 

### 3.2. Negative Relationships of Religiosity and Spirituality with Health-Related Outcomes of Adolescents with Chronic Illnesses

Just as religiosity and spirituality are potentially beneficial resources for adolescents living with a chronic medical condition, they might also be detrimental. In this context, the way one conceives God and one’s relationship with God might inhibit the process of *coping* with the disease by leading the individual to feel punished, abandoned, or spiritually discontented. Some qualitative studies reported that young patients became angry with the divine, although temporarily, as they perceived their condition as unfair [42]. 

These sensations, in turn, seemed to have a negative impact on the *psychosocial adjustment* and *well-being* of adolescents with chronic illnesses. Reynolds and colleagues [55] found that negative spiritual coping (spiritual discontentment, negative reappraisals of God’s power, and demonic reappraisals) was related to more externalizing problems (hyperactivity, aggression, and conduct problems) among adolescents suffering from diabetes and cystic fibrosis. Negative spiritual coping was related to more internalizing problems (depression and anxiety) only among adolescents with cystic fibrosis, suggesting that this association may be specific to adolescents with more severe conditions. Interestingly, in this sample, older adolescents were reported to use more negative spiritual coping than younger ones, which might be due to their advanced cognitive abilities to reflect on the enduring repercussions of their medical condition. Additionally, in a longitudinal study with the same sample, Reynolds and colleagues [56] reported reciprocal relationships between negative spiritual coping and conduct problems only at baseline for patients with diabetes and for those with cystic fibrosis. However, this maladaptive strategy predicted positive spiritual coping over time, which might indicate that experiencing spiritual struggles might push individuals to redirect their relationship with the sacred. 

*Adherence to medical treatment* and *physical health* have been observed to be negatively affected by the way adolescents experienced their relationships with the divine. Regarding this, Grossoehme and colleagues [51] reported that spiritual struggle (not asking for God’s help and questioning God’s love) was associated with lower perceived utility of airway clearance treatment among adolescents with cystic fibrosis, which, in turn, was linked to lower intentions to perform it and adherence to the therapy. Regarding *physical health*, Grossoehme and colleagues [52] reported an association between worse clinical trajectory, that is, faster retrospective pulmonary function decline, and the use of negative religious coping among young patients with cystic fibrosis. 

Religiosity and spirituality might also negatively impact the *quality of life* of adolescents with chronic illnesses; in this regard, Taha and colleagues [60] proposed a complex picture. In particular, among a sample of adolescents with spina bifida, they tested a moderated mediation model in which distress related to physical symptoms (symptom distress) was linked to quality of life by means of depressive symptoms, while spirituality (a global scale encompassing spiritual beliefs and perceived support) moderated the relationship between depressive symptoms and quality of life. In particular, adolescents with high levels of spirituality and low/moderate levels of depressive symptoms had a lower life quality compared with those adolescents with low/moderate levels of both depressive symptoms and spirituality. To note, those with severe depressive symptoms had the lowest life quality, regardless of their levels of spirituality. With regard to these findings, scholars speculated that depressive symptoms were more relevant than spirituality for the quality of life of adolescents with spina bifida. However, they suggested that adolescents with higher spirituality may, in any case, experience hopelessness or a lack of existential support. Such a condition might intensify the adverse influences of depressive symptoms on the quality of life of adolescents with this chronic illness. 

## 4. Discussion

In the current work, 20 articles were scoped in order to report on the state of the literature exploring the impact of religious and spiritual beliefs, thoughts, and practices on the health-related outcomes of adolescents suffering from a chronic disease. Although this is not the first contribution on the topic, the present synthesis differs from previous literature and meta-analytic reviews [10,11,12] because it contemplated several chronic medical conditions, referred to different dimensions of adolescents’ religiosity and spirituality, focused exclusively on the adolescent years, and explored mechanisms accounting for the associations between religious and spiritual factors and the health-related outcomes of adolescents with chronic illnesses.

In general, the findings supported the recourse of adolescents to facets of their religiosity and spirituality in their experience of illness and aligned with works carried out among adult samples [62]. In detail, the results showed that religiosity and spirituality might have both beneficial and deleterious effects on ill adolescents’ process of coping with their enduring medical condition, psychosocial adjustment, mental and physical health, quality of life, and adherence to medical treatment. However, a closer examination of the results might unveil relevant and interesting observations.

Firstly, religiosity and spirituality are neither good nor bad per se; rather, it is the way that young patients approach them that might make these two dimensions helpful or detrimental in the context of chronic illness [63]. Regarding this, the illness narratives of young patients highlighted that, regardless of the type of disorder and religious affiliation, adolescents who placed trust in God, reinforced their relationship with God through prayers or religious services/activities (e.g., reading Holy Scriptures), and saw God as merciful and loving, felt less isolated, and more relieved and comforted, and had a positive outlook on their medical situation [42,45,49]. Quantitative studies, similarly, showed that adolescents with chronic illnesses who used positive religious/spiritual coping strategies (e.g., reframing one’s situation from a spiritual perspective) had fewer internalizing and externalizing problems [55], adhered to medical treatment [51], and reported better health outcomes [57]. Conversely, those adolescents with chronic illnesses who felt abandoned by God, considered their medical condition as a punishment, or did not ask for God’s help reported greater levels of maladjustment [55], less adherence to the therapies [51], and worse clinical trajectories [52]. These findings align with a significant body of empirical studies showing that religious and spiritual coping might help and harm people who are facing stressful life events (for review, [64]). 

Secondly, religiosity and spirituality might support young patients in unique ways [45]. In particular, spirituality seemed to be a factor promoting resilience, because it made adolescents with chronic illnesses experience connectedness with others, nature, arts, and creative activities; these connections, in turn, helped adolescents with a chronic condition to relax and stop thinking of their problems. Religiosity, instead, was more concerned with having a personal relationship with God and being committed to one’s beliefs, which provided young patients with hope, comfort, encouragement, and with the strength to persevere while confronting difficult moments. Such qualitative findings seemed to support that evidence underscoring that religiosity and spirituality might be independent or combined resources affecting healthy adolescents’ outcomes and thriving [65,66].

Thirdly, beyond personal and individual experiences of religiosity and spirituality, some works in this scoping review highlighted that religious communities [43,58] and spiritual anchors (e.g., parents; [60]) played a relevant role in the lives of adolescents struggling with a chronic illness, as they supported them spiritually and emotionally and gave them the strength to fight against their condition. Thus, these findings echoed those theoretical arguments and the empirical evidence revealing that religiosity and spirituality provide adolescents with social contexts where they might come into contact with sources of inspiration and webs of support, as well as have trustworthy relationships that favor their healthy development [35].

Fourthly, possible underlying mechanisms that could explain the relations between religiosity and spirituality and the health-related outcomes of adolescents with chronic illnesses emerged. In particular, perceptions of the utility of therapies, treatment-supporting norms from friends, and intentions to perform the treatment seemed to explain the pathways from spiritual factors to adherence to medical treatments [51]. In addition, meaning and peace accounted for the associations between features of one’s religiosity/spirituality and anxiety, depression, and fatigue [50]. Finally, an optimistic attribution style explained the linkages between positive spiritual coping and both internalizing and externalizing problems [55]. The scoped studies seemed to stress that religiosity and spirituality might act as meaning systems offering the motivation to follow medical prescriptions and to interpret one’s illness-related problems, as well as a sense of purpose in life [8].

Finally, some cultural/ethnic, socioeconomic status, and age-related differences impacted the way adolescents used religious and spiritual resources to deal with their chronic medical conditions. In terms of cultural/ethnic variances, although, in the scoped works, believers from any religious background (e.g., Muslims, Christians, and Buddhists) were found to draw solace and comfort from their religiosity and spirituality, Muslim adolescents seemed to turn more to their faith compared with their Christian counterparts [42]. This might be due to the fact that religious identity is more salient in Muslim culture compared with Western culture and is thus more central in several aspects of individuals’ lives [67], including illness. 

With regard to socioeconomic status, it seemed that disadvantaged adolescents were likely to rely more often on their faith in order to cope with their illness [53]. At first glance, it could be argued that religion is the only resource people with lower education and income might hold on to when going through difficult times [68], but it is worth noticing that the nexus between religion and socioeconomic status might be influenced by a number of historical and cultural factors, which are also country-specific [69]. In any case, this issue deserves more investigation among young people.

Regarding age-related variances, it seemed that, on the one hand, older adolescents were more intrinsically motivated to explore and to turn to their religion to find meaning and purpose than younger ones, who tended to more passively accept the observances of their families, as emerged in the study of Atkin and Ahmad [42] among Muslims. On the other hand, older adolescents with a chronic condition were also reported to adopt more often negative spiritual coping activities compared with younger ones [55]. Altogether, this evidence might be explained considering the advances in cognitive functioning that characterize the different phases of adolescence. In particular, older adolescents are more able to think abstractly, to process information more accurately, and to reflect on different aspects of the self and of their culture [70]. These abilities might translate into a more active and individual exploration of religious beliefs and values [40], which might lead to questioning God’s love and struggling spiritually at times [55], especially in the context of chronic diseases, whose enduring and irreversible consequences are clearer to older adolescents [31]. 

In conclusion, it emerged that the associations between spiritual coping and adjustment might vary by disease. For example, Reynolds and colleagues [55] highlighted that negative spiritual coping was related to depression and anxiety only among adolescents with cystic fibrosis, and not among those with diabetes, suggesting that this association may be specific to adolescents with more severe conditions. Moreover, they showed that, in adolescents with cystic fibrosis, an optimistic attributional style did not mediate the effects of positive spiritual coping on adjustment, such as in adolescents with diabetes, suggesting that other explanatory mechanisms may intervene for adolescents with cystic fibrosis. Thus, it might be a case that spiritual coping–psychosocial adjustment relationships, as well as the effectiveness of mediators in such associations, might depend on the severity of disease and on its implications for young patients’ lives. 

### 4.1. Implications

The present scoping review has implications for both research and practice. With regard to research, the studies analyzed in this synthesis underlined the need to adopt both qualitative and quantitative approaches when exploring the connections between the features of religiosity and spirituality of adolescents with chronic illnesses and their influences on young patients’ psychosocial adjustment and mental/physical health. Indeed, a qualitative approach seemed to be a useful means to gain a deeper insight into how young patients define these two constructs (e.g., [45]) and into the reasons leading them to rely on their faith and religious/spiritual beliefs, thoughts, practices, and communities to go through disease-related circumstances (e.g., [42,43]). 

Regarding the second approach, quantitative studies were suitable for testing theoretical models revealing the underlying mechanisms connecting religiosity/spirituality to various health-related outcomes of adolescents with chronic illnesses (e.g., [51]). In addition, the few longitudinal works helped to ascertain temporal order and causality between religiosity/spirituality and young patients’ health (e.g., [57]) or psychosocial adjustments [56], as well as reciprocal links between health, adjustment, and adolescents’ religious/spiritual coping (e.g., [48,56]). 

Regarding practical implications, the scoped studies offered important clinical information that might be useful for healthcare professionals. In detail, as the religiosity and spirituality of adolescents with chronic illnesses have been reported to impact young patients’ mental/physical health and adherence to medical treatments, clinicians should identify the role of such dimensions by using validated measures [11], such as those screening negative and positive spiritual coping [71]. This might allow detecting spiritual struggles and addressing them accurately, by asking for counseling from spiritual/religious leaders.

Furthermore, doctors, nurses, and professionals should provide young patients with spiritual care by showing empathy, humanity, kindness, and, eventually, by sharing thoughts or starting conversations on existential/spiritual questions (e.g., [43]). To do so properly, healthcare professionals should be trained to become aware of the importance of religiosity and spirituality in adolescents’ lives and of the differences between theirs and young patients’ religious/spiritual traditions and cultures in order to avoid prejudice and lack of understanding [41]. Finally, it could be fruitful for clinicians to consider age-related differences in the way that adolescents live their religiosity and spirituality and talk to them about these issues accordingly [40]. To sum up, healthcare professionals have a relevant role in young patients’ existence as they might help them to carry the burden of their medical condition and manage their chronic illness during the pivotal time of adolescence. Notably, this seems to imply that they become more and more aware of the uniqueness of their patients and seek to understand their world and feelings [72], which might also include both a positive personal relationship with the divine and spiritual struggles.

### 4.2. Limitations and Future Research

Although this scoping review reported some interesting results, it had limitations. First, shortcomings related to the research criteria adopted to carry out this synthesis of the literature should be considered. Indeed, querying only three databases (*Scopus*, *Web of Science*, and *PubMed*) and accepting only studies published in English might have hindered the possibility to gather other relevant articles. In light of this, future scoping reviews should search other databases and include works written in other languages. 

Second, the limits of the scoped studies should be taken into account as well. To begin, only a few studies compared religiosity and spirituality between groups of adolescents with different chronic diseases [55,56]. Thus, future works should include more diverse samples in order to investigate whether the influences of religious and spiritual dimensions on the health-related outcomes of adolescents with chronic illnesses might vary as a function of their various medical conditions. In addition, only a few studies, which were all cross-sectional, tested potential mechanisms explaining the associations between religious and spiritual factors and the adolescents’ psychosocial adjustment (e.g., [55]) and health (i.e., adherence to therapies; [51]). Future studies should adopt longitudinal designs to firmly establish such mediation processes. Furthermore, the reviewed contributions did not take into account several other potential mechanisms (e.g., religious social support) by which religiosity and spirituality might impact the health-related outcomes of adolescents with chronic illnesses [8,73]. 

Another limitation concerns the fact that most of the quantitative studies originated from Western countries and that most of their participants were from a Christian background. It would be interesting to replicate these studies elsewhere in order to understand to what extent these findings can be generalized to other populations with different religious/spiritual traditions. Additionally, studies analyzing the associations between the features of religiosity and spirituality and health-related outcomes of adolescents with chronic illnesses mostly adopted measures and scales screening negative and positive spiritual coping strategies. In light of this, future studies should opt for instruments detecting religious behaviors (e.g., frequency of prayers and church/temple attendance) or capturing adherence to one’s religious values, beliefs, and traditions in order to ascertain whether religious practices and commitments have a part in the process of coping with chronic disease among adolescents. Finally, measures used were self-reported for the most part, which might lead to social desirability. Thus, future studies should adopt less direct types of assessment. 

Despite these limitations, the current scoping review reported findings from some qualitative studies [49,60] that allowed the exploration of the experiences and practices of adolescents from spiritual/religious backgrounds other than a Christian one. In particular, some of those works permitted the detection of cultural and ethnic variances in the ways adolescents with different religious beliefs turned to their faith and spiritual resources to deal with their chronic medical condition (e.g., [42]).

## 5. Conclusions

Adolescence is a crucial time of life during which several challenges arise and changes at the biological, psychological, and social levels occur [1]. Chronic illnesses might interfere with the processes of growth as they impose lifestyles on adolescents that limit their social relationships and impact their psychosocial well-being and life quality [4]. The studies scoped in the current review highlighted that features of adolescents’ religiosity and spirituality might help young patients go through their medical condition by providing them with positive outlooks, by buffering the onset of both internalizing and externalizing problems, and by promoting adherence to medical therapies. However, the reviewed works also showed the negative effects of religiosity and spirituality on the health-related outcomes of adolescents with chronic illnesses, which are likely to occur when adolescents perceive their situations as a punishment from God, question God’s love, and lose their sense of meaning in life. Despite these interesting findings, this synthesis of the literature stressed that studies examining the role of religiosity and spirituality in the lives of adolescents suffering from a chronic disease are still scarce, especially those adopting a developmental perspective and exploring the mechanisms connecting these two dimensions with the health-related outcomes of these adolescents. However, evidence so far has reported that religiosity and spirituality might provide young patients with comfort and a sense of purpose, thus indicating that, in the context of chronic illness, healthcare professionals should consider adolescents’ religious and spiritual dimensions as potential resources helping them to live through the difficult times of their medical condition.

## Figures and Tables

**Figure 1 ijerph-19-13172-f001:**
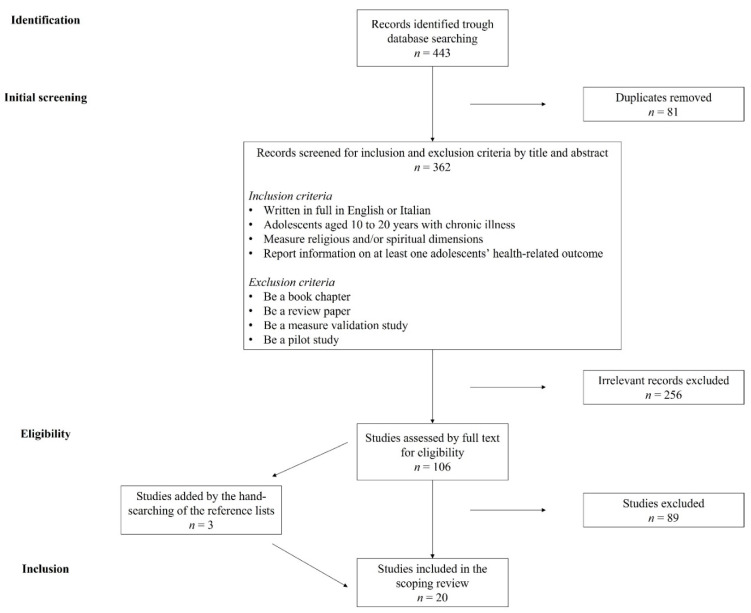
PRISMA diagram illustrating the literature selection process.

**Table 1 ijerph-19-13172-t001:** Main characteristics and findings of the studies included in this scoping review.

Authors	Participants	Age Range (years)	Country	Characterization of the Sample	Instruments/Measures	Main Results
Atkin and Ahmad, 2001 [42]	*N* = 51 Female = 27	10–19	UK	Disease: Sickle cell disorder (SCD) or thalassemia major (TM) Religious Affiliation: Islamic or Christian	Semi-structured interviews (administered twice over a 6-month period) to explore the strategies and resources young people used to cope with their disorders by encouraging them to talk about their illness within their social context (i.e., family relationships, life transitions, and social networks).	Age, gender, and ethnicity influenced how spiritual belief was utilized as a resource. Youth with TM were mostly Islamic and generally saw Allah as a source of strength. Their resentment toward the “unfairness” of the illness was transitory. Most of them prayed on a regular basis for a cure (<13 years) or for gaining strength (older participants). In general, younger children and girls passively accepted Islam (i.e., they followed their family), while older boys (> 16 years) were more active believers (e.g., they often read the Koran). Accepting one’s fate and “passing tests” sent by God helped them to make sense of their condition. Youth with SCD were mostly Christian and adopted religion as a coping resource less frequently. Most of them prayed to God for need (e.g., children < 12 for relief), but sometimes had the feeling of being ignored. They felt that, if they were generally good, they would not have a crisis (reward vs. punishment). Age seemed to diminish the importance of religion.
Alvarenga et al., 2021 [43]	*N* = 35 Female = 17	7–18	Brazil	Disease: Cancer, cystic fibrosis, or type 1 diabetes mellitus. Religious Affiliation: 14 Evangelical; 13 Catholic; 1 Umbanda (Afro-Brazilian religion); 1 Spiritism; 1 No religion, but spiritual (believes in something); 0 No religion and not spiritual (does not believe in anything); 5 Atheist (does not believe in God)	Individual audio-recorded interviews with photo-elicitation centered on the experience of the disease and the role of religion/spirituality in the life journey, life meaning, religious/spiritual beliefs, and resources used to cope with the disease (religious/spiritual beliefs, family, friends, and health professionals).	Children and adolescents with chronic illnesses had five spiritual needs while in the hospital: (1) Need to integrate meaning and purpose in life, for instance, by believing that their disease was part of a plan of a benevolent God who wanted them to mature through their illness; (2) Need to sustain hope, especially about their future; faith is an element that helps to promote hope; (3) Need for expression of faith and to follow religious practices: they believed in a benevolent God who intervened in adverse or near-death situations, to keep them alive or heal them; they described the religious community as a source of support and comfort; (4) Need for comfort at the end of life by believing in a life after death (e.g., the existence of hell and paradise); (5) Need to connect with family and friends, as a source of faith, peace, and support while dealing with the illness and the finitude of life. They also found comfort in believing that, after their own deaths, they will be reunited with their deceased family members in a good place. Additionally, participants conveyed that not enough spiritual care was offered in the hospital due to the professionals’ lack of time and difficulty in dialogue on this subject.
Bernstein et al., 2013 [44]	*N* = 45 (n_HIV_ = 19) Female = 28	12–21 *M* = 17.2 *SD* = 2.2	USA	Disease: HIV Religious affiliation: 15 Baptist; 5 Church of Christ; 1 Lutheran; 1 Methodist; 7 Non-denominational Christian; 1 Orthodox Church; 1 Other Protestant; 2 Pentecostal; 3 Roman Catholic; 1 Southern Baptist; 1 Undesignated; 2 Other; 3 None	A survey packet containing the measures of spirituality/religiosity, quality of life measures, acceptance of spiritual discussions, and general demographic variables	Teens with HIV were more likely to endorse wanting their doctors to pray with them, feeling ‘‘God’s presence’’, being ‘‘part of a larger force’’, and feeling that ‘‘God had abandoned them’’ than their counterparts without HIV.
Clayton-Jones et al., 2016 [45]	*N* = 9 Female = 6	15–18 *M* = 16.2	USA	Disease: Sickle cell disease (SCD) Religious affiliation: Baptist, Catholic, Pentecostal, and Presbyterian	A qualitative descriptive design was used. Two semi-structured interviews were conducted with adolescents.	Teens expressed that they drew from spirituality and religiosity to cope with SCD, but in different ways. Spirituality was seen to enhance their sense of connectedness with one another, nature, and the arts, which helped them to feel better, find purpose, and transcend their condition. Religiosity was seen in terms of a connection with God, also via the reading of scriptural metanarratives, which lead adolescents to gain new outlooks on their illness, strength, hope for the future, and frameworks for decision-making and reflection.
Cotton et al., 2012 [46]	*N* = 151 Female = 91	11–19 *M* = 15.8 *SD* = 1.8	USA	Disease: Asthma Religious affiliation: 97 Protestant; 16 Catholic; 1 Jewish; 35 No Preference; 2 Other	Demographic data and adolescents’ religious preferences were collected via patient interviews. Asthma severity at the time of the study was collected via a clinical provider according to the National Heart, Lung, and Blood Institute criteria (National Heart Lung and Blood Institute 2007).	African American race/ethnicity and having a religious preference were related to higher levels of spirituality/religiosity (S/R), including positive religious coping. With increasing clinical severity, adolescents’ preferences for including S/R in the medical setting grew.
Cotton et al., 2009 [47]	*N* = 154 (n_IBD_= 66) Female = 74	11–19 *M* = 15.1 *SD* = 2.0	USA	Disease: Inflammatory bowel disease (IBD) Religious affiliation: -	Questionnaires were administered to measure spiritual (religious and existential) well-being, depression, emotional functioning, and health-related quality of life, as well as demographics, disease status, and their interactions.	Most adolescents believed that a Higher Power loved and cared about them, and more than half reported that their relationship with a Higher Power contributed to their well-being. Adolescents with and without IBD showed similar levels of both existential and religious well-being. However, the disease status moderated the relationship between spiritual well-being and mental health outcomes. Indeed, (a) the positive relationship between existential well-being and emotional functioning and (b) the inverse relationship between religious well-being and depressive symptoms were both stronger for adolescents with IBD than for their healthy peers.
D’Angelo et al., 2021 [48]	*N* = 79 Female = 43	12–18 *M* = 14.7 *SD* = 1.8	USA	Disease: Cystic fibrosis Religious affiliation: 67 Christian, 4 Other, 8 No affiliation	Questionnaires assessing secular and religious/spiritual coping styles at two timepoints (18 months apart, on average). Health indicators, including pulmonary functioning, nutritional status, and days hospitalized, were obtained from medical records.	Poorer pulmonary functioning predicted higher levels of positive religious/spiritual coping, suggesting the resilience of adolescents with cystic fibrosis. More frequent hospitalizations, instead, may inhibit the use of adaptive coping strategies over time.
Elissa et al., 2018 [49]	*N* = 9 Female = 4	8–18	Palestine	Disease: Congenital heart disease (CHD) Religious affiliation: Muslims	An inductive qualitative descriptive design with face-to-face interviews at home was applied. The interview guide included the following main questions: “Can you describe your CHD and how do you think it affects your daily life?”, “What is a typical day like for you right now?”, and “On a typical day, what sorts of things do you do that might set you apart from your friends?”	All children believed that everything in the universe, including health or illness, was controlled by God’s will, and, as consequence, it should be tolerated rather than objected. They adopted a sense of fatality about illness and reliance on God for managing the disease and controlling community pressure. Some participants also held the belief that God could heal illness, so they regularly engaged in religious practices, including reading from the Holy Qur’an and praying at a mosque as a means of coping and searching for support and hope.
Grossoehme et al., 2020 [50]	*N* =126 Female = 72	14–21 *M* = 16.9 *SD* = 1.9	USA	Disease: Cancer Religious affiliation: 24 Agnostic/atheist/none, 90 Christian, 1 Hindu, 1 Jehovah’s Witness, 1 Jewish, 6 LDS/Mormon, 3 Missing	Sociodemographic data (i.e., age, sex, race, ethnicity, education, and household income); time since diagnosis, treatment status, study site; the importance of religion and spirituality to participants; religiousness/spirituality (i.e., feeling God’s presence, daily prayer, religious service attendance, being very religious, and being very spiritual); spiritual well-being (meaning/ peace and faith); and anxiety, depressive symptoms, fatigue, and pain interference.	Through a higher sense of meaning and peace: (a) experiencing God’s presence every day was indirectly related to anxiety, depressive symptoms, and fatigue; (b) being highly religious was indirectly related to anxiety, depressive symptoms, and fatigue; (c) being highly spiritual was indirectly associated with anxiety and depression. No links between spiritual scales and pain interference were found.
Grossoehme et al., 2016 [51]	*N* = 45 Female = 27	11–19 *M* = 13.8 *SD* = 2.2	USA	Disease: Cystic fibrosis Religious affiliation: 19 Nondenominational Christian, 10 Protestant, 6 Roman Catholic, 6 None, 3 Other, 1 Did not disclose	Psychosocial, spiritual coping, treatment attitude (utility), subjective norms, sanctification of the body, self-efficacy, treatment intentions, and treatment adherence.	Lower levels of “spiritual struggle” (i.e., not asking for God’s help or questioning God’s love) and higher levels of “engaged spirituality” (i.e., positive religious coping, collaboration with God to solve problems, turning problems to God, or viewing one’s body as sacred) predicted treatment attitude (utility) as well as subjective behavioral norms, which, in combination with self-efficacy, predicted treatment intentions. Additionally, treatment intentions predicted adherence to airway clearing.
Grossoehme et al., 2013 [52]	*N* = 28 Female = 9	11–18 *M* = 13.5	USA	Disease: Cystic fibrosis Religious affiliation: -	Religious coping (“negative religious coping styles” and “pleading style of religious coping for control”).	Adolescents who experienced lung function decline more quickly were more likely to use pleading or negative religious coping styles. A negative correlation existed between certain religious coping styles and longitudinal changes in lung functioning. Positive rates of change in lung functioning were related to less pleading. The probability of using any religious coping was lower for slower pulmonary function decline, but, when compared with pleading, the probability of engaging in any negative religious coping did not decrease as quickly. Hence, even when adolescents’ lung function was above the normal range of that of their healthier counterparts, they still used negative religious coping.
Landolt et al., 2002 [53]	*N* = 179 (n_cancer_ = 26, n_diabetes_ = 48) Female = 69	*M* = 10.2 *SD* = 2.3	Switzerland	Disease: Cancer or type I diabetes mellitus Religious affiliation: -	Coping (i.e., active coping, distraction, avoidance, support seeking, and religiosity), functional status, and socioeconomic status	Patients used a wide range of coping strategies, but those of lower socioeconomic status turned to religious coping strategies far more frequently than their counterparts.
Lyon et al., 2014 [54]	*N* = 38 Female = 23	14–21 *M* = 16.6 *SD* = 2.3	USA	Disease: HIV Religious affiliation: -	Spiritual well-being (faith and meaning/peace), psychological adjustment (depression and anxiety), and health-related quality of life.	Higher adolescents’ spiritual well-being was related to lower depression, lower anxiety, and greater life quality.
Reynolds et al., 2013 [55]	*N* = 128 Female = 59	12–18 *M* = 14.7 *SD* = 1.8	USA	Disease: Cystic fibrosis or type 1 diabetes Religious affiliation: Predominantly Christian	Demographics, positive spiritual coping (i.e., seeking spiritual support or collaboration from God, as well as benevolent religious reappraisals) vs. negative spiritual coping (i.e., spiritual discontentment, negative reappraisals of God’s powers, or demonic reappraisals); attributional style; and adolescents’ adjustment (internalizing and externalizing problems).	Positive spiritual coping was related to less internalizing and externalizing problems. Negative spiritual coping was associated with more externalizing problems, and solely for teens with cystic fibrosis, internalizing problems as well. Optimistic attributions mediated the effects of positive spiritual coping among diabetic teens.
Reynolds et al., 2014 [56]	Same as in the previous study (at baseline) *N* = 87 (at follow-up)	12–18 at baseline *M* = 14.7 *SD* = 1.8 Follow-up age was ~2 years after baseline *M* = 1.78 *SD* = 0.80	USA	Disease: Cystic fibrosis or type 1 diabetes Religious affiliation: 11% no religious affiliation, 78% Protestant, 8% Catholic, 3% other.	Spiritual coping and adjustment, adolescent adjustment (2 years apart).	Over time, less negative spiritual coping and depressive symptoms were predicted by positive spiritual coping, whereas more positive spiritual coping was predicted by negative spiritual coping. Higher levels of negative spiritual coping and conduct problems over time were predicted by depressive symptoms. The results did not vary by disease.
Reynolds et al., 2014 [57]	*N* = 46 Female = 23	12–18 *M* = 14.7 *SD* = 1.9	USA	Disease: Cystic fibrosis Religious affiliation: Predominately Christian	Spiritual coping, secular coping, pulmonary function, BMI percentile, hospitalizations, baseline medical complications, and demographics.	Positive spiritual coping was linked to a slower decrease in pulmonary function, stable vs. declining nutritional status, and fewer days spent in the hospital over the course of five years. Negative spiritual coping was linked to a higher BMI percentile at baseline, but not to long-term health outcomes.
Silveira and Neves, 2019 [58]	*N* = 35	12–18	Brazil	Disease: Children and adolescents who need special healthcare services Religious affiliation: -	A qualitative, descriptive, and exploratory study. Semi-structured interviews were conducted with adolescents, followed by the construction of genograms and ecomaps.	Some adolescents saw the church as a source of spiritual support that enabled them to cope with the challenges created by their medical conditions. The search for spirituality as emotional support and a source of strength aided in the socialization process, as the church and the youth group became part of the adolescent’s social network.
Taha et al., 2020 [59]	*N* = 58 Female = 29	13–20 *M* = 16.2 *SD* = 2.2	USA	Disease: Spina bifida Religious affiliation: 32 Protestant; 20 Catholic; 2 Agnostic; 2 Atheist; 2Other	Spirituality; depression and quality of life; and distress.	Depressive symptoms fully mediated the association between symptom distress and quality of life, and higher levels of spirituality moderated the relationship between depressive symptoms and quality of life. Adolescents with more severe symptoms (i.e., Welch’s shunt status, level of lesion, and ambulation status) had higher spirituality. Contrary to predictions, when depression symptoms were mild to moderate, adolescents with higher levels of spirituality had a lower quality of life.
Thanattheerakul et al., 2020 [60]	*N* = 17 Female = 6	10–18 *M* = 13.5 *SD* = 2.09	Thailand	Disease: Cancer (35.3%), bone and joint (29.4%), neurology and urology (both 11.8%), and endocrine and immunology (both 5.85%). Religious affiliation: Buddhist	Data were collected by using a questionnaire, in-depth interviews with questions adapted from the Spiritual Assessment Scale (SAS; O’Brien, 2014), and non-participant observation	Children reported that, when they were sick, their mothers and other family members served as their spiritual anchors, and their physicians and sacred spirituals were significant as well. They afforded them the inner strength to battle their illness and live their lives. In addition, interviews showed that (a) children believed that doing good deeds could protect them, especially during illness; (b) spiritual practices (e.g., prayer, requesting blessings on sacred things) increased when they were ill and had credit to bring inner peace and relief.
Zehnder et al., 2006 [61]	*N* = 161 (n_disease_ = 60) Female = 63 (n_disease_ = 24)	6–15 *M* = 10.0 *SD* = 2.3	Switzerland	Disease: 25 type 1 diabetes, 23 cancer, and 12 epilepsy. Religious affiliation: -	Coping (i.e., active coping, distraction, avoidance, support seeking, and religious coping, namely asking for God’s help and praying to God for comfort), child post-traumatic stress reactions, behavioral problems, socio-economic status, functional status, and preceding life events.	Religious coping reduced post-traumatic stress symptoms among (injured and) newly diagnosed children with a chronic disease after 1 year.

## Data Availability

Not applicable.

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
