# Peer review of "How Do Religiosity and Spirituality Associate with Health-Related Outcomes of Adolescents with Chronic Illnesses? A Scoping Review"

_ijerph, 2022, doi:10.3390/ijerph192013172_

Round 1
Reviewer 1 Report
This review is very well written, comprehensive and balanced. In particular, I appreciated this point, "Firstly, religiosity and spirituality are not per se neither good nor bad, rather it is the way young patients approach them that might make these two dimensions helpful or detrimental in the context of chronic illness [61]." The implications for research and practice are well done and useful.
There were a few instances of awkwardness in English grammar and style. I put in suggested word changes on the pdf. The suggestions are highlighted on the following pages 1,2,3,5,6,7,9,15,16,17
The phrase "adolescents' outcomes" does not make sense really. It is not clear to this reader what that means.
I recommend changing that phrase to "health-related outcomes" on pages 193, 210 and in the Prisma chart. All of the outcomes that were identified are health-related. All of the studies, by taking teens with chronic conditions as their subjects, make them health-related studies. Changing that phrase to health-related outcomes would make the criteria more understandable to readers. And it would be more specific because the outcomes were health-related.
Following on that, I would also suggest a change in title to "How do Religiosity and Spirituality Relate to Health-Related Outcomes of Adolescents with Chronic Illnesses?"
Putting the term "health-related" in the title may also increase interest in this paper as people are interested in R&S and health.
Also, adolescents with chronic illnesses is a preferred description over chronically ill adolescents - people first!

Author Response
Manuscript Title: “How do Religiosity and Spirituality Associate with Health-Related Outcomes of Adolescents with Chronic Illnesses? A scoping review”
Dear Editor,
thank you very much for offering us the opportunity to submit a revision of the above-titled manuscript. We would like to express our sincerest gratitude to the Reviewers for the constructive and thoughtful feedbacks. As you can see, we reviewed the manuscript in line with reviewers’ suggestions, which helped us improve it. Firstly, we changed the title in “How do Religiosity and Spirituality Associate with Health-Related Outcomes of Adolescents with Chronic Illnesses? A scoping review”. Secondly, we better outlined the relations between religiosity and spirituality in the introduction and expanded the implications for practitioners in the “Implications” section. Thirdly, we corrected the mistakes in the text. Fourthly, we reformulated all those expressions that might be confusing, such “adolescents’ outcomes”, or inappropriate, such as “chronically ill adolescents”. Finally, we added some relevant references. In this letter, we have detailed how each editorial feedback has been addressed. In the manuscript, all changes were tracked.
Comments of, and responses to, Reviewer 1
This review is very well written, comprehensive and balanced. In particular, I appreciated this point, "Firstly, religiosity and spirituality are not per se neither good nor bad, rather it is the way young patients approach them that might make these two dimensions helpful or detrimental in the context of chronic illness [61]." The implications for research and practice are well done and useful.
Reply: Thank you very much for your appreciation.
(Query 1) There were a few instances of awkwardness in English grammar and style. I put in suggested word changes on the pdf. The suggestions are highlighted on the following pages 1,2,3,5,6,7,9,15,16,17
Reply: Thank you very much for noticing them. We followed your suggestions and corrected all the mistakes.
(Query 2) The phrase "adolescents' outcomes" does not make sense really. It is not clear to this reader what that means. I recommend changing that phrase to "health-related outcomes" on pages 193, 210 and in the Prisma chart. All of the outcomes that were identified are health-related. All of the studies, by taking teens with chronic conditions as their subjects, make them health-related studies. Changing that phrase to health-related outcomes would make the criteria more understandable to readers. And it would be more specific because the outcomes were health-related. Following on that, I would also suggest a change in title to "How do Religiosity and Spirituality Relate to Health-Related Outcomes of Adolescents with Chronic Illnesses?". Putting the term "health-related" in the title may also increase interest in this paper as people are interested in R&S and health.
Reply: Thank you for noticing this. We changed this expression across the entire text and in the title as well. Now the title is “How do Religiosity and Spirituality Associate with Health-Related Outcomes of Adolescents with Chronic Illnesses? A scoping review”.
(Query 3) Also, adolescents with chronic illnesses is a preferred description over chronically ill adolescents - people first!
Reply: Thank you. We do agree and changed this expression accordingly across the text and in the title.
Reviewer 2 Report
Overall, this is a well done scoping review informing about the current understanding of how R/S constructs relate to the illness experience of adolescents with chronic illness. The researchers presented a nice introduction arguing the relevance of this review, defining the central concepts, and explaining their assumptions, central hypothesis/starting point and then delineated nicely in the method how they went about. Although they acknowledged limitations, I think they captured a decent part of the literature by searching three data bases systematically. They represented their inclusion/exclusion criteria clearly and depicted the process accurately in a PRISMA diagram. The tables provided a rich source to orient the reader to the main findings that entered the analysis. The results were then summarized in a meaningful way presenting a nuanced perspective of findings that corroborate or are contradictory/non conclusive. The team followed up with a meaningful discussion of the findings from this body of literature leading to cogent conclusions. The work that has been accomplished is helpful for healthcare providers who work with chronically ill adolescents because these children see healthcare providers more often than do children without a chronic illness. Compliance/adherence to chronic illness self management is always an important issue. So healthcare providers may wonder if they can engage adolescents' spirituality/religiosity (S/R) in a way that helps them cope with the burden of the illness and with self-management. To me, this may be an aspect that is alluded to but could maybe be addressed more clearly in the discussion/conclusion. I'd like to thank the authors for their careful and well balanced review that is practice relevant.
While the appreciation for the solid work of the team is unquestioned, allow me to briefly mention a few minor observations:
1) p.2 line 51 should say "clarify" instead of clarified
2) The S/R construct. I appreciate your careful analysis and deeper dive into the role that different subdimensions of the S/R construct play conceptually and later in the text in the lives of adolescents with chronic illness. I appreciate your summary of the pertinent literature citing key authors like Pargament, Hill, Miller, Thoresen, etc. and capturing some of the scholarly conceptual discussion. On p.1 line 68, you are insinuating that science strives to provide one clear conceptualization for the constructs. I would argue that this will never be possible because of the diversity of opinions, beliefs, and experiences of these constructs. I also don't think that it is productive to conceptualize spirituality as "privatized, experience-oriented religion" (line 79). While the latter is one facet of it, we cannot reduce spirituality to just that. Likewise, religiosity is not only institutional religion - what does that even mean? Attendance of worship? Conforming to certain creeds? This is a polarization in order to explain what is different between S and R. I don't find these definitions helpful in clarifying anything. Allow me to suggest Zinnbauer et al's (1997) attempt to to clarify descriptively because most people can identify with one of the following four categories: Spiritual AND religious, spiritual but not religious, religious but not spiritual. neither spiritual nor religious. Furthermore, he suggests conceptualizations of that suggest S & R are completely overlapping, partially overlapping, or two distinct non-overlapping terms. Again, people then decide what they identify with rather than researchers giving one authorative perspective. I find that view more accurately reflecting the diversity of perspectives that we find in different populations. So my suggestion is to discuss with the team whether it would be more productive to work descriptively with the variety of ideas rather than some of the confusing definitions that researchers have come up with and that often clarify very little. Zinnbauer, B. J., Pargament, K. I., Cole, B., Rye, M. S., Butter, E. M., & Belavich, T. G. (1997). Spirituality and religion: Unfuzzying the fuzzy. Journal for the Scientific Study of Religion, 36, 549-64. Let me however also say that as you work through the findings from the 20 articles, you bring out the richness of the concepts and it becomes clear how meaningful the S/R construct is for review.
3) p. 2 line 78 when you cite source 18, does it really refer to a plural or is it divine being/higher power?
Author Response
Manuscript Title: “How do Religiosity and Spirituality Associate with Health-Related Outcomes of Adolescents with Chronic Illnesses? A scoping review”
Dear Editor,
thank you very much for offering us the opportunity to submit a revision of the above-titled manuscript. We would like to express our sincerest gratitude to the Reviewers for the constructive and thoughtful feedbacks. As you can see, we reviewed the manuscript in line with reviewers’ suggestions, which helped us improve it. Firstly, we changed the title in “How do Religiosity and Spirituality Associate with Health-Related Outcomes of Adolescents with Chronic Illnesses? A scoping review”. Secondly, we better outlined the relations between religiosity and spirituality in the introduction and expanded the implications for practitioners in the “Implications” section. Thirdly, we corrected the mistakes in the text. Fourthly, we reformulated all those expressions that might be confusing, such “adolescents’ outcomes”, or inappropriate, such as “chronically ill adolescents”. Finally, we added some relevant references. In this letter, we have detailed how each editorial feedback has been addressed. In the manuscript, all changes were tracked.
Comments of, and responses to, Reviewer 2
Overall, this is a well done scoping review informing about the current understanding of how R/S constructs relate to the illness experience of adolescents with chronic illness. The researchers presented a nice introduction arguing the relevance of this review, defining the central concepts, and explaining their assumptions, central hypothesis/starting point and then delineated nicely in the method how they went about. Although they acknowledged limitations, I think they captured a decent part of the literature by searching three data bases systematically. They represented their inclusion/exclusion criteria clearly and depicted the process accurately in a PRISMA diagram. The tables provided a rich source to orient the reader to the main findings that entered the analysis. The results were then summarized in a meaningful way presenting a nuanced perspective of findings that corroborate or are contradictory/non conclusive. The team followed up with a meaningful discussion of the findings from this body of literature leading to cogent conclusions. The work that has been accomplished is helpful for healthcare providers who work with chronically ill adolescents because these children see healthcare providers more often than do children without a chronic illness. Compliance/adherence to chronic illness self-management is always an important issue. So healthcare providers may wonder if they can engage adolescents' spirituality/religiosity (S/R) in a way that helps them cope with the burden of the illness and with self-management.
Reply: Thank you very much for your appreciation.
(Query 1) To me, this may be an aspect that is alluded to but could maybe be addressed more clearly in the discussion/conclusion. I'd like to thank the authors for their careful and well balanced review that is practice relevant.
Reply: Thank you for your suggestion. Now we expanded the “Implications” section in order to better address how professionals might help adolescents cope with the burden of their illness. Now you can read:
“To sum up, healthcare professionals have a relevant role in young patients’ existences as they might help them carry the burden of their medical condition and manage their chronic illness during the pivotal time of adolescence. Notably, this seems to imply that they become more and more aware of the uniqueness of their patients and seek to un-derstand their world and feelings [72], which might also include both a positive per-sonal relationship with the divine and spiritual struggles.”.
While the appreciation for the solid work of the team is unquestioned, allow me to briefly mention a few minor observations:
(Query 2) p.2 line 51 should say "clarify" instead of clarified
Reply: Thank you for noticing this. We corrected the mistake.
(Query 3) The S/R construct. I appreciate your careful analysis and deeper dive into the role that different subdimensions of the S/R construct play conceptually and later in the text in the lives of adolescents with chronic illness. I appreciate your summary of the pertinent literature citing key authors like Pargament, Hill, Miller, Thoresen, etc. and capturing some of the scholarly conceptual discussion. On p.1 line 68, you are insinuating that science strives to provide one clear conceptualization for the constructs. I would argue that this will never be possible because of the diversity of opinions, beliefs, and experiences of these constructs. I also don't think that it is productive to conceptualize spirituality as "privatized, experience-oriented religion" (line 79). While the latter is one facet of it, we cannot reduce spirituality to just that. Likewise, religiosity is not only institutional religion - what does that even mean? Attendance of worship? Conforming to certain creeds? This is a polarization in order to explain what is different between S and R. I don't find these definitions helpful in clarifying anything. Allow me to suggest Zinnbauer et al's (1997) attempt to to clarify descriptively because most people can identify with one of the following four categories: Spiritual AND religious, spiritual but not religious, religious but not spiritual. neither spiritual nor religious. Furthermore, he suggests conceptualizations of that suggest S & R are completely overlapping, partially overlapping, or two distinct non-overlapping terms. Again, people then decide what they identify with rather than researchers giving one authorative perspective. I find that view more accurately reflecting the diversity of perspectives that we find in different populations.
So my suggestion is to discuss with the team whether it would be more productive to work descriptively with the variety of ideas rather than some of the confusing definitions that researchers have come up with and that often clarify very little. Zinnbauer, B. J., Pargament, K. I., Cole, B., Rye, M. S., Butter, E. M., & Belavich, T. G. (1997). Spirituality and religion: Unfuzzying the fuzzy. Journal for the Scientific Study of Religion, 36, 549-64. Let me however also say that as you work through the findings from the 20 articles, you bring out the richness of the concepts and it becomes clear how meaningful the S/R construct is for review.
Reply: Thank you very much for your useful suggestions that we believed helped us corroborate our theoretical argumentation. We integrated your suggestions by also considering the reference you mentioned and changed part of that paragraph accordingly:
“With regard to their reciprocal relations, religiosity and spirituality have been often regarded as different but overlapping constructs [15] sharing the reference to the sacred [17], which encompasses the divine, the transcendent, or any aspect of life that indi-viduals perceive as imbued with a spiritual/divine character [18]. Interestingly, spiritu-ality has been considered as a phenomenon pertaining to all individuals regardless of their religious affiliation [19], and religiosity as one of the possible contexts where one’s spiritual inclinations might be expressed [20,21]. In any case, it is worth noticing that in their lived experiences of religiosity and spirituality individuals might self-describe as spiritual and religious, spiritual but not religious, religious but not spiritual, neither re-ligious nor spiritual, according to the meanings attributed to these terms”
(Query 4) p. 2 line 78 when you cite source 18, does it really refer to a plural or is it divine being/higher power?
Reply: Thank you for noticing this. We reformulated this part in line with your suggestions:
“With regard to their reciprocal relations, religiosity and spirituality have been often regarded as different but overlapping constructs [15] sharing the reference to the sacred [17], which encompasses the divine, the transcendent, or any aspect of life that individuals perceive as imbued with a spiritual/divine character [18]”
Reviewer 3 Report
This is a well written article, well researched, and an important contribution. I would check whether it is valuable to add some of the citations for chronicity including: Estroff; Manderson et al; Mattingly.
Author Response
Manuscript Title: “How do Religiosity and Spirituality Associate with Health-Related Outcomes of Adolescents with Chronic Illnesses? A scoping review”
Dear Editor,
thank you very much for offering us the opportunity to submit a revision of the above-titled manuscript. We would like to express our sincerest gratitude to the Reviewers for the constructive and thoughtful feedbacks. As you can see, we reviewed the manuscript in line with reviewers’ suggestions, which helped us improve it. Firstly, we changed the title in “How do Religiosity and Spirituality Associate with Health-Related Outcomes of Adolescents with Chronic Illnesses? A scoping review”. Secondly, we better outlined the relations between religiosity and spirituality in the introduction and expanded the implications for practitioners in the “Implications” section. Thirdly, we corrected the mistakes in the text. Fourthly, we reformulated all those expressions that might be confusing, such “adolescents’ outcomes”, or inappropriate, such as “chronically ill adolescents”. Finally, we added some relevant references. In this letter, we have detailed how each editorial feedback has been addressed. In the manuscript, all changes were tracked.
Comments of, and responses to, Reviewer 3
This is a well written article, well researched, and an important contribution.
Reply: Thank you for your appreciation.
(Query 1) I would check whether it is valuable to add some of the citations for chronicity including: Estroff; Manderson et al; Mattingly.
Reply: Thank you for your suggestion. We searched for some articles by scholars you mentioned and decided to add this one to our work (see line 524).
Unantenne, N., Warren, N., Canaway, R., & Manderson, L. (2013). The Strength to Cope: Spirituality and Faith in Chronic Disease. J Relig Health 52, 1147–1161. doi.org/10.1007/s10943-011-9554-9